# Evaluation of the Influence of Short Tourniquet Ischemia on Tissue Oxygen Saturation and Skin Temperature Using Two Portable Imaging Modalities

**DOI:** 10.3390/jcm11175240

**Published:** 2022-09-05

**Authors:** Wibke Müller-Seubert, Helen Herold, Stephanie Graf, Ingo Ludolph, Raymund E. Horch

**Affiliations:** Department of Plastic and Hand Surgery and Laboratory for Tissue Engineering and Regenerative Medicine, University Hospital Erlangen, Friedrich Alexander University Erlangen-Nürnberg (FAU), 91054 Erlangen, Germany

**Keywords:** tourniquet ischemia, tissue oxygen saturation, skin temperature

## Abstract

Background: The exact influence of tourniquet ischemia on a treated extremity remains unclear. Methods: Twenty patients received an operation on one hand under tourniquet ischemia. Twenty healthy volunteers received 10 min of tourniquet ischemia on one of their arms. Measurements of tissue oxygen saturation using near-infrared reflectance-based imaging and skin temperature of the dorsum of the hand were performed at five different timepoints (t0 was performed just before the application of the tourniquet ischemia, t1 directly after the application of the tourniquet ischemia, t2 before the release of the ischemia, t3 directly after the release of the ischemia, and t4 on the following day). Results: In both groups, tissue oxygen saturation dropped after the application of the tourniquet ischemia compared to t0 and increased after the release of the tourniquet ischemia. In the patient group, tissue oxygen saturation at t4 was higher compared to t0; in contrast, the level of tissue oxygen saturation in the participant group dropped slightly at t4 compared to t0. The measured skin temperature in the patient group showed an increase during the observation period, while it continuously decreased in the group of healthy participants. Conclusions: Short-term ischemia did not appear to permanently restrict perfusion in this study design. The non-invasive imaging modalities used were easy to handle and allowed repetitive measurement.

## 1. Introduction

Using an upper arm tourniquet that enables a bleeding-free situs is the state of the art in the context of hand surgery [1]. However, tourniquet-related injuries are the result of compression by the tourniquet and ischemia distal to the cuff [2]. Even though tourniquet use seems not to influence the appearance of complications following surgical treatment, for example, an open lower-extremity fracture [3], it still has an effect on the treated limb. While compression might be the main cause of nerve injury, resulting in axonal degeneration, the duration of the ischemia might influence muscular injury [2,4,5]. The upper extremity seems to be more susceptible to nerve damage than the lower extremity [4]. Furthermore, the release of the tourniquet might result in an ischemia reperfusion injury. On the one hand, reperfusion after ischemia is necessary to prevent irreversible cellular damage. On the other hand, the restoration of blood flow can augment the damage caused by ischemia. The severity of the cellular damage depends on the duration of the ischemia [6]. Ischemia and reperfusion are characterized by the increased adhesiveness, entrapment and activation of leucocytes [7]. The application of a tourniquet seems not to completely occlude plasma blood flow [8]. The combination of muscle ischemia, edema and microvascular congestion can end in post-tourniquet syndrome, which is characterized by stiffness, pallor, weakness without paralysis and subjective numbness of the extremity without objective anesthesia [4]. To avoid this worst possible consequence, the use of tourniquet ischemia needs to be considered carefully and its mechanism of action investigated.

The aim of the present study was to measure the changes in blood flow to the hand in the context of ischemia and reperfusion after application of a tourniquet. Near-infrared imaging (NIRI) and thermographic imaging were used to measure the changes in temperature and tissue oxygen saturation. These imaging modalities are already used in reconstructive plastic surgery and in the context of solid organ transplantation [9,10,11,12]. NIRI has been described in plastic surgery to monitor free flaps postoperatively by measuring tissue oxygen saturation [13]. Thermographic imaging has already been investigated in the context of assessing the perfusion of random pattern flaps or mastectomy skin flaps and in the context of wound healing [14,15,16].

## 2. Methods

The study was approved by the institutional ethics committee (310_19 B9). Twenty patients received an operation on one of their hands and tourniquet ischemia was applied. Furthermore, 20 healthy participants, who voluntarily participated in the study, received 10 min of tourniquet ischemia in one of their arms. Both patients and healthy participants were prospectively included in the study. In both groups, the tourniquet pressure was set at 300 mmHg, which is the standardized pressure for adults in our clinic and, after exsanguination, measurements of tissue oxygen saturation and skin temperature of the dorsum of the hand were performed at five different timepoints. The first measurement (t0) was performed just before application of the tourniquet ischemia, the second (t1) directly after application of the tourniquet ischemia, the third (t2) before release of the ischemia, the fourth (t3) directly after release of the ischemia and the last (t4) was performed on the following day (Figure 1). The number of healthy participants was chosen according to the number of patients, and both subgroups were equal in size and similar in gender ratio and age composition.

Three measuring points (MP) were defined at the dorsum of the hand: at the head of the third metacarpal bone (MP1), the middle of the third metacarpal bone (MP2) and the base of the third metacarpal bone (MP3). The mean value of these three measuring points was used for statistics.

As previously described, near-infrared reflectance-based imaging was performed using Snapshot NIR^®^ (KENT Imaging Inc., Calgary, AB, Canada) to measure tissue oxygenation in superficial tissue (Figure 2) [17]. This is a technique whereby light is transmitted to the skin. Briefly, light is selectively absorbed or reflected in order to calculate the percentage of oxygenated and deoxygenated hemoglobin. The tissue oxygen perfusion can be assessed by calculating this ratio [18]. More precisely, two convergent laser points integrated in the device overlap at a distance of about 32 cm determined the standardized distance. The near-infrared light that is transmitted onto the skin surface is reflected off the blood within the tissue. The ratio of oxygenated to deoxygenated blood and therefore the viability can be measured because of the wavelength-dependent difference of oxygenated and deoxygenated light absorption of hemoglobin. In conclusion, well-perfused skin has a higher percentage of oxygenated hemoglobin than poorly perfused skin [18].

Thermal imaging was performed using either a smartphone-compatible thermographic camera (FLIR ONE Pro, FLIR Systems, Inc., Wilsonville, OR, USA) (Figure 3) or a smartphone with an integrated FLIR-thermographic camera (Cat S61—Smartphone, Caterpillar Inc., Deerfield, IL, USA), which uses the same long-wave infrared sensor. The provided effective temperature range was set from −20 °C to 400 °C. It has a resolution of 0.1 °C and a sensitivity that detects temperature differences down to 70 mK. A photograph can be merged with a thermal image by specific image processing. The images were not taken from a previously determined distance. However, the automatic calibration of the device at regular intervals as well as the automatic readjustment of the measuring distance by the corresponding Flir One R—App speak for the fact that the Flir One R Pro—measuring device is a standardized measurement [19].

The statistics were performed using GraphPad Prism (GraphPad Software Inc., San Diego, CA, USA). For every patient/participant, mean values for each timepoint were calculated. The mean values for each timepoint either of the patient or of the participant group were compared to t0. If the values were normally distributed, the Repeated-Measures ANOVA Test was used; if not, the Friedman Test in combination with the Dunn’s Multiple Comparisons Test was used. Subgroup analysis was performed for gender and age (subgroup 1: 18 to 56 years; subgroup 2: older than 56 years). Age subgroups were divided in such a way that each subgroup consisted of the same number of individuals (n = 10).

## 3. Results

Between April 2019 and December 2021, 20 patients (10 male, 10 female) were included in this study with a mean age of 57.85 years (±10.52 years). All operations (A1 pulley release, decompression of the median nerve) were performed under local anesthesia including 10 right and 10 left hands. The mean time for ischemia was 14.35 min (±4.13 min; range: 7–24 min).

Furthermore, between September 2020 and November 2021, 20 healthy participants (10 male, 10 female) were recruited with a mean age of 49.3 years (±22.69 years). The participants received tourniquet ischemia to the left (n = 2) or right arm (n = 18) with a pressure of 300 mmHg for 10 min.


Oxygen saturation in patients:


After the application of the tourniquet ischemia, there was a statistically significant drop in oxygen saturation (t1: *p* = 0.04; t2: *p* < 0.0001) compared to t0 (Figure 4 and Figure 5). At t3, an increase in oxygen saturation was measured, with the mean value being higher than the baseline value. At t4, the measured oxygen saturation was slightly reduced compared to t3, but still higher compared to t0. The differences at timepoints t3 and t4 compared to t0 are not statistically significant (t3: *p* = 0.23; t4: *p* = 0.36).


Skin temperature in patients:


The measured skin temperature (Figure 6 and Figure 7) increased continuously both after the application of the tourniquet ischemia and after release of the tourniquet ischemia as well as on the first postoperative day compared to t0. The differences at timepoints t1 (*p* = 0.03), t3 (*p* < 0.01) and t4 (*p* < 0.01) were, in contrast to the value at t2 (*p* = 0.18), statistically significant.


Oxygen saturation in healthy participants:


Tissue oxygen saturation (Figure 4) decreased after the application of the tourniquet ischemia statistically significantly compared to t0 (t1: *p* = 0.01; t2: *p* < 0.0001). After opening the tourniquet, tissue oxygen saturation increased statistically significantly above the level of t0 (t3: *p* = 0.02). At t4, tissue oxygen saturation decreased slightly compared to the beginning (t4: *p* > 0.99).


Skin temperature in healthy participants:


The measured temperature (Figure 6) decreased continuously over the observation period, but the differences compared to t0 were not statistically significant (t1: *p* = 0.85; t2: *p* = 0.99; t3: *p* = 0.72; t4: *p* = 0.50).

In the patient group, the subgroup analysis did not show a statistically significant difference in tissue oxygen saturation or skin temperature at any time point, neither by age nor by gender.

In the subgroup analysis of the healthy participant group, there was no difference in tissue oxygen saturation or skin temperature between the male and the female subgroups. In contrast, tissue oxygen saturation was, in the older subgroup 2 (20.17%), statistically significantly lower compared to the younger subgroup 1 (48.87%) at t4 (*p* = 0.005). Furthermore, skin temperature was in the older subgroup 2 (27.60 °C) statistically significantly lower compared to the younger subgroup 1 (32.22 °C) at t2 (*p* = 0.0006).

## 4. Discussion

In this study, the influence of tourniquet ischemia on the perfusion of the affected limb was examined by measuring tissue oxygen saturation and skin temperature.

Both patients who received surgery on the corresponding hand and healthy participants were examined with a longer follow-up of 24 h compared to other studies [20,21]. Interestingly, the measured skin temperature in the patient group showed an increase during the observation period, while it continuously decreased in the group of the healthy participants. However, that certain differences were measured between these two groups is not surprising, as the patient group had an additional influencing factor: the manipulation of the operation itself.

It has already been shown that tourniquet ischemia influences temperature not only of the treated extremity, but also of the whole body. Akata et al. showed that the tourniquet ischemia of one leg resulted in an increase in body temperature followed by a decrease in body temperature after the release of the ischemia [22]. They assumed that the restriction of metabolic heat to the thermal core compartment caused an increase in body core temperature; the following vasodilatation resulted in higher skin surface temperature measured at the fingertip. Conversely, the opening of the tourniquet led to a backflow of hypothermic venous blood from the tourniquet limb and caused a decrease in body core temperature and skin surface temperature due to the stopping of thermoregulatory vasodilatation [22]. Radowsky et al. used a large animal hind limb model for critical ischemia with both a tourniquet and a direct vessel occlusion technique [23]. The evaluation of perfusion was performed using infrared imaging. They demonstrated a more progressive and almost linear decrease in temperature over the duration of ischemia, but the values of temperature returned to baseline levels by 30 min postreperfusion. These results are similar to those of our participant group.

In both groups, tissue oxygen saturation dropped after the application of the tourniquet ischemia compared to t0 and increased after the release of the tourniquet ischemia. In both groups, the tissue oxygen saturation decreased from timepoint t3 to t4 but resulted in higher levels compared to t0 in the patient group, while measured values at t4 in the participant group dropped to the level of t0. The decrease in saturation during ischemia was expected. The results of this study using two different imaging modalities confirmed the subsequent rapid increase in saturation to baseline or even beyond as has already been described in other studies.

Babilas et al. measured the transcutaneous partial oxygen pressure before, during and after tourniquet ischemia using luminescence lifetime imaging and the application of transparent sensor foils [20]. Reperfusion was observed for 20 min after tourniquet release. They showed that partial oxygen pressure levels increased to baseline values after the release of the tourniquet.

As in our study, where we observed an increase in tissue oxygen saturation after the release of the tourniquet ischemia compared to the baseline values, Lin et al. measured this reactive hypersaturation after the release of the tourniquet ischemia in patients undergoing ankle surgery using NIRI [21]. In contrast to our study, they did not measure tissue oxygen saturation one day after the tourniquet ischemia. This phenomenon of reactive hyperemia has already been described elsewhere [24]. It is arterial in nature because a higher increase in oxyhemoglobin and oxymyoglobin could be measured in the tissue compared to the decrease in deoxyhemoglobin and myoglobin during recovery from ischemia [24]. Kim et al. measured an increase in tissue oxygen saturation compared to the baseline values using NIRI after 2 h of tourniquet ischemia in a rat limb model, while the values after 3 h of ischemia were lower compared to the baseline values [25]. These results indicate that the duration of ischemia has an influence on a possible hyperemic response. Tujjar et al. showed that the hyperemic peak of the treated upper extremity measured using NIRI was also lower in patients with a longer duration of ischemia [26]. Reactive hyperemia in healthy volunteers was shown as a significant increase in the regional oxygen saturation above the initial values after release of an upper arm tourniquet ischemia [27]. In rat paws, the maximal hyperemic response increased as the duration of occlusion increased. Measurement was performed using Laser Doppler [28].

The necessity of tourniquet ischemia should be evaluated carefully as ischemia and the following reperfusion cause complex processes that can end in muscle degeneration and loss of function [29]. The local injury might be aggravated by an inflammatory response in the ischemic tissue that occurs after reperfusion [30]. Recovery from ischemia depends predominantly on the microcirculation of the tissue [23]. Since microcirculation involves the smallest vessels, it is responsible for tissue oxygenation [23,31]. Ischemia affects the endothelial cells, causes aggregation of the red blood cells and impairment of the arterioles are impaired, which is why they can no longer regulate perfusion [23]. Furthermore, ischemia activates leukocytes that build reactive oxygen species that impair microcirculation [23]. Due to the changes in microcirculation, a longer duration of ischemia results in increasing vascular permeability to plasma proteins and progressive interstitial edema [30,32]. The no-reflow phenomenon might occur, when due to far-advanced muscle ischemia, the nutrient vessels are closed and blood flow does not return to all areas [29,32].

While the short duration of ischemia in our study does not seem to have left any permanent visible damage, harmful side effects of tourniquet ischemia have already been described. Total knee arthroplasty with a tourniquet is associated with an increased risk of severe adverse events, pain, and a marginally longer hospital stay compared to the same operation without tourniquet ischemia. The only finding in favor of tourniquet use is a shorter time in theatre [33]. Tourniquet use causes ischemia and cell damage as measured by metabolites such as Pyruvate and Lactate in total knee arthroplasty. Significant ischemia can be detected in the affected limb over a period of 3 h after release of the tourniquet, but all marker levels are normalized after 5 h [34].

The lower values of skin temperature and tissue oxygen saturation at two timepoints in the healthy participant group might be explained as increasing age is an independent risk factor for the development of atherosclerosis [35] and due to that impairs microcirculation [36].

In contrast to others [26], we evaluated the influence of the tourniquet ischemia under local anesthesia to avoid the possible influence of plexus anesthesia on tissue oxygen saturation. As already described, the use of plexus anesthesia or the perioperative placement of a plexus catheter leads to an increase in local blood flow to the ipsilateral upper extremity and to vasodilatation [37,38,39].

Our study evaluated the perfusion of tissue using two non-invasive imaging modalities, whose application was simple, easy to repeat and easy to handle with small portable devices and without exposing the participants to harmful radiation. Perfusion was determined indirectly by measuring temperature and tissue oxygen saturation. More precise parameters of perfusion, for example, postischemic skeletal muscle blood flow can be estimated using contrast enhanced MRI [40]. However, this is much more time-consuming, is dependent on expensive equipment and does not allow bedside examinations. Laser Doppler flowmetry, which has been used in other studies [41,42], can record the tissue perfusion continuously, while the measurements in our study were only taken at specific timepoints. Indocyaningreen angiography is used predominantly to assess tissue perfusion in flap surgery [43,44]. The differences between well-perfused, mal-perfused and non-perfused areas of the flaps are clearly shown. However, the disadvantages of this imaging modality are the necessity of applying a dye and the expensive equipment.

One limitation of the study was that the participant and patient groups could be compared with each other to a limited extent. The patients who had surgery received a dressing or splint that was still in place at the time of the last measurement and could have warmed the treated hand, while the participants did not receive a dressing. In addition, it can be assumed that the patients probably did not use their operated hand during the observation period, whereas the participants may have done exactly the opposite, such as during sport. This different behavior might have influenced the measured tissue oxygen saturation and skin temperature. Furthermore, perfusion was evaluated only indirectly via the measurement parameters of oxygen saturation and temperature. Therefore, small measurement inaccuracies must be assumed. For example, thermal imaging using FLIR ONE in a rat perforator model showed a sensitivity, specificity, and accuracy with a temperature difference of 80.8, 83.6, and 82 percent, respectively [45]. The assessment of burn wounds showed that the FLIR ONE was highly reliable, but with moderate validity [46].

## 5. Conclusions

This study aimed to investigate the influence of short-term ischemia on the perfusion of an affected limb indirectly by measuring tissue oxygen saturation and temperature. The non-invasive imaging modalities used were easy to handle and allowed for repetitive measurement. Short-term ischemia does not appear to restrict perfusion permanently in this study design with a longer follow-up of 24 h. Further studies would be useful in order to investigate the influence of a longer ischemia duration in a larger patient collective.

## Figures and Tables

**Figure 1 jcm-11-05240-f001:**
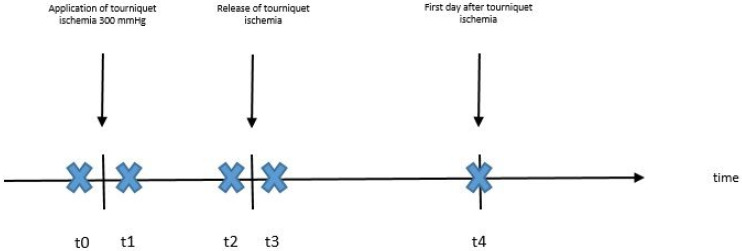
Study design and measuring timepoints (t0–t4).

**Figure 2 jcm-11-05240-f002:**
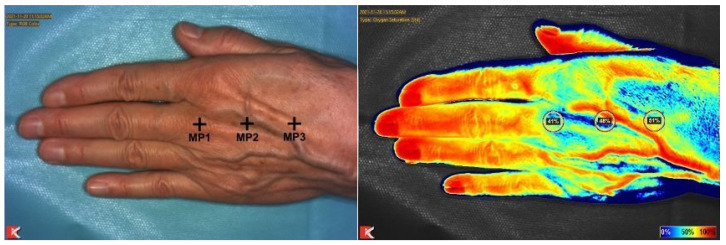
Tissue oxygen saturation measured at three different measuring points (MP) with NIRI.

**Figure 3 jcm-11-05240-f003:**
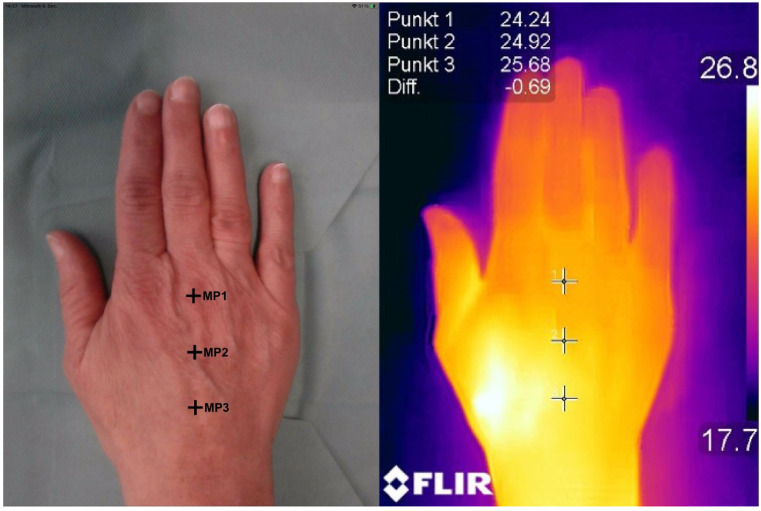
Skin temperature measured at three different measuring points (MP) with thermal imaging.

**Figure 4 jcm-11-05240-f004:**
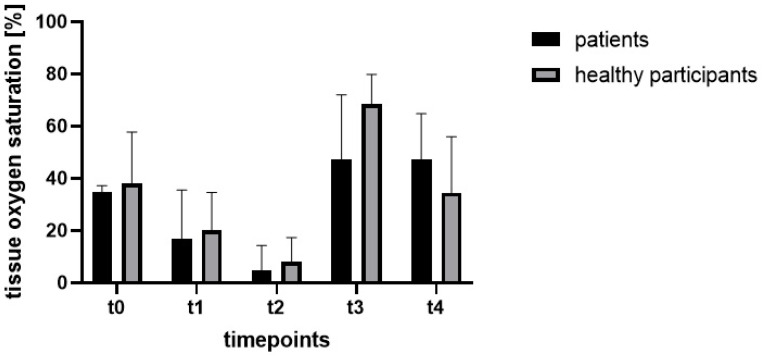
Mean values of measured tissue oxygen saturation at different timepoints.

**Figure 5 jcm-11-05240-f005:**
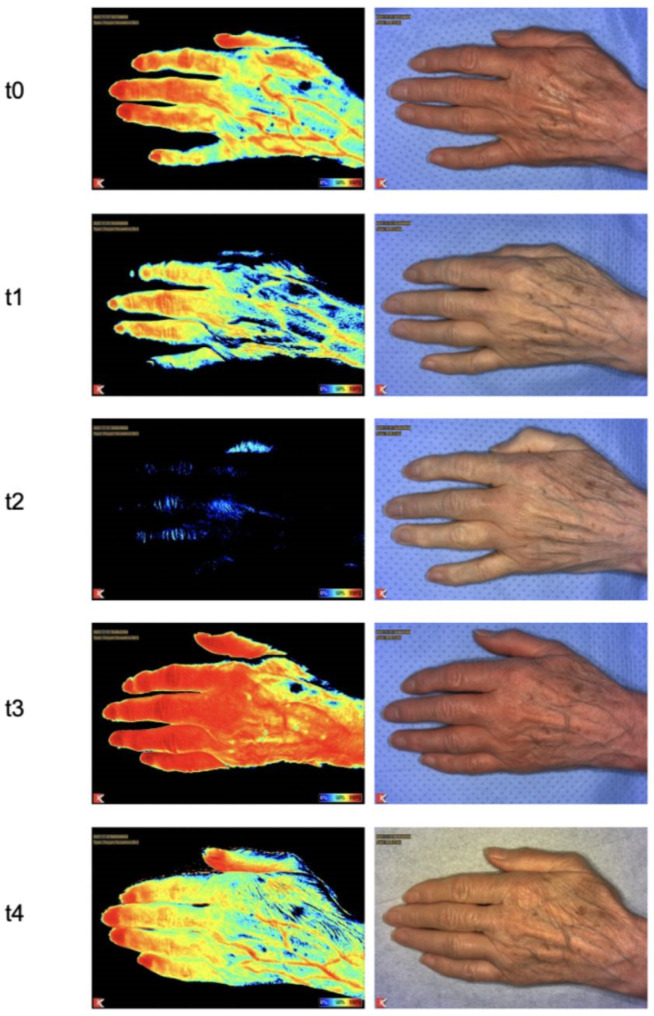
Tissue oxygen saturation of a patient’s hand at different timepoints.

**Figure 6 jcm-11-05240-f006:**
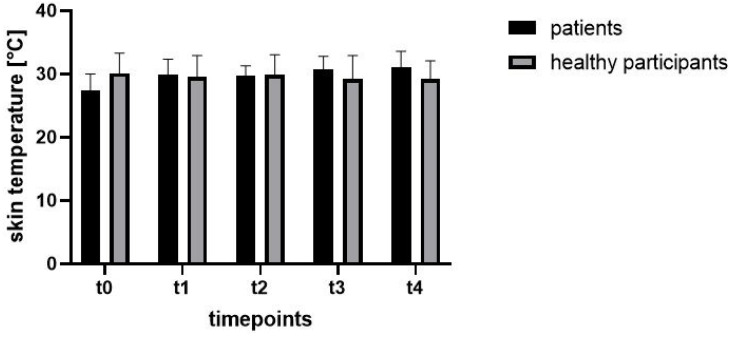
Mean values of measured skin temperature at different timepoints.

**Figure 7 jcm-11-05240-f007:**
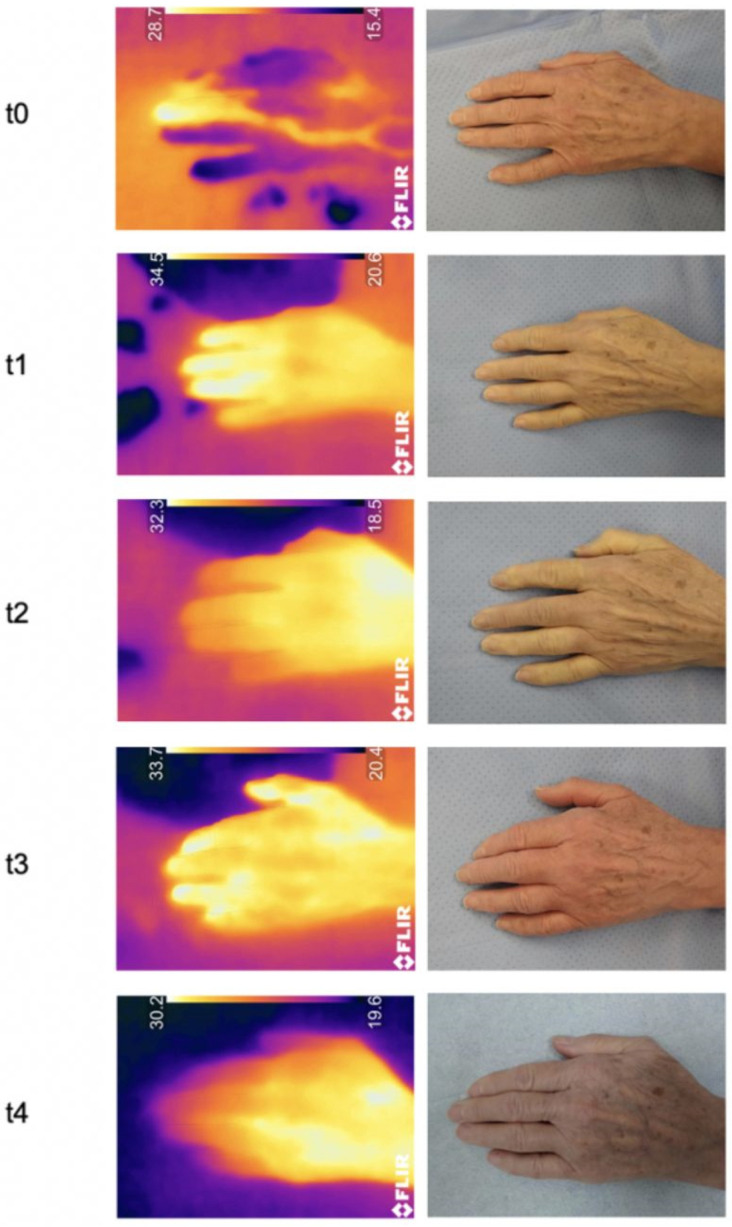
Skin temperature of a patient´s hand at different timepoints.

## Data Availability

The data presented in this study are available on request from the corresponding author.

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
