# Peer review of "Evaluation of the Influence of Short Tourniquet Ischemia on Tissue Oxygen Saturation and Skin Temperature Using Two Portable Imaging Modalities"

_jcm, 2022, doi:10.3390/jcm11175240_

Round 1
Reviewer 1 Report
The data may be simple, but could be important for the future researches for scientist and clinicians in regards to the continuing application of tourniquet and the transitional standard of performing hand and extremity surgery without tourniquet.
There several points to be considered by the authors:
Regarding the Abstract,
1. The term “participants” needs to be mentioned more specifically as we could only assume that they were healthy subjects who voluntarily participated in the research.
2. The equipment used to measure oxygen saturation needs to be mentioned in the Methods section. Consequently, the results of both devices are also to be described in the Results section.
In the body of the manuscript, some other points need clarification:
3. How was the decision on the subject number made?
4. How was the recruitment of the subjects, especially patient subjects? As the patients were recruited in April 2019 to December 2021, while the healthy subjects were enrolled in September 2020 - November 2021, the authors need to clarify on how the nature of the study was conducted.
It could be assumed that the patient subjects were recruited retrospectively; on the other hand, the healthy subjects were recruited prospectively.
5. It is suggested that the authors may not mention the weakness of the study as one. Besides splinting/ dressing and activities, it could be predicted that there are more potential weakness to be identified and stated, especially the nature of the methodology.
Author Response
Dear Reviewer 1,
We appreciate your suggestions and tried our best to improve the manuscript according to the given suggestions.
In the following paragraphs we tried to address all notes from the reviewers in a point-by-point manner (the changes we made to the manuscript are highlighted in red letters in the revised version):
Point 1:” The term “participants” needs to be mentioned more specifically as we could only assume that they were healthy subjects who voluntarily participated in the research.”
We titled the "participants" more specifically and explained their participation in more detail in the methods section
Point 2: “The equipment used to measure oxygen saturation needs to be mentioned in the Methods section. Consequently, the results of both devices are also to be described in the Results section.“
We added near infrared spectroscopy in this section.
Point 3: “How was the decision on the subject number made?”
Thank you for this comment, we have explained that 20 patients and 20 healthy participants were included in order to obtain groups that are as comparable as possible.
Point 4. “How was the recruitment of the subjects, especially patient subjects? As the patients were recruited in April 2019 to December 2021, while the healthy subjects were enrolled in September 2020 - November 2021, the authors need to clarify on how the nature of the study was conducted. It could be assumed that the patient subjects were recruited retrospectively; on the other hand, the healthy subjects were recruited prospectively. “
Both patients and healthy participants were included prospectively, we added this information in the method´s section.
Point 5: “It is suggested that the authors may not mention the weakness of the study as one. Besides splinting/ dressing and activities, it could be predicted that there are more potential weakness to be identified and stated, especially the nature of the methodology. “
We added further points in the limitations´ section such as the indirect measurement and the reliability and validity of the thermal imaging methods.
Reviewer 2 Report
The unique advantage of the IR technique in biomedicine is that the subject is not exposed to dangerous radiation during the clinical assessment
of the patients. In this study, the authors specifically used the infrared camera model FLIR One, which was launched to the market for the non-invasive, non-contact monitoring of skin surfaces. This smartphone-based IR thermal monitoring device belongs to a new generation thermal camera and is being actively used for biomedical purposes nowadays.
Although it is useful for measurements in the field where traditional methods may be impractical, the NIR thermal camera system may have several limitations. To help our readers judge the quantitative results of the study fairly, the reviewer recommends the authors state the validity and reliability of the FLIR One system, at least shortly.
FLIR One is a fixed focal distance thermal camera, which
tends to result in low accuracy if used in hand-held mode or for targets from about 45 cm and further. Did the authors capture the surface temperature at a fixed distance? If yes, what was the distance between the camera and the skin surface? Please specify.
The NIR reflectance-based imaging system (Snapshot NIR) is an emerging technique to evaluate tissue perfusion and oxygenation using relevant wavelengths in the VIS-NIR region. Since this technology is relatively new to the readers in the surgical field, it is better for the authors to briefly mention the measuring device's basic technology and mode of action.
Author Response
Dear Reviewer 2,
We appreciate your suggestions and tried our best to improve the manuscript according to the given suggestions.
In the following paragraphs we tried to address all notes from the reviewers in a point-by-point manner (the changes we made to the manuscript are highlighted in red letters in the revised version):
Point 1: “The unique advantage of the IR technique in biomedicine is that the subject is not exposed to dangerous radiation during the clinical assessment of the patients. In this study, the authors specifically used the infrared camera model FLIR One, which was launched to the market for the non-invasive, non-contact monitoring of skin surfaces. This smartphone-based IR thermal monitoring device belongs to a new generation thermal camera and is being actively used for biomedical purposes nowadays.
Although it is useful for measurements in the field where traditional methods may be impractical, the NIR thermal camera system may have several limitations. To help our readers judge the quantitative results of the study fairly, the reviewer recommends the authors state the validity and reliability of the FLIR One system, at least shortly. “
We added that the imaging methods we have used don´t expose the participants to harmful radiation. Furthermore, we added studies showing the validity and reliability of the flir one system.
Point 2: “FLIR One is a fixed focal distance thermal camera, which tends to result in low accuracy if used in hand-held mode or for targets from about 45 cm and further. Did the authors capture the surface temperature at a fixed distance? If yes, what was the distance between the camera and the skin surface? Please specify.”
We did not determine a specific distance but due to the automatic calibration we assume the measured values are standardized.
Point 3:”The NIR reflectance-based imaging system (Snapshot NIR) is an emerging technique to evaluate tissue perfusion and oxygenation using relevant wavelengths in the VIS-NIR region. Since this technology is relatively new to the readers in the surgical field, it is better for the authors to briefly mention the measuring device's basic technology and mode of action.”
We explained this technology more precisely in the method´s section.